# Moles of Molecules against *Mycobacterium abscessus*: A Review of Current Research

Mario Cocorullo [ID], Christian Bettoni, Sara Foiadelli and Giovanni Stelitano *[ID]

Department of Biology and Biotechnology "Lazzaro Spallanzani", University of Pavia, Via A. Ferrata 9, 27100 Pavia, Italy; mario.cocorullo01@universitadipavia.it (M.C.); christian.bettoni01@universitadipavia.it (C.B.); sara.foiadelli01@universitadipavia.it (S.F.)
* Correspondence: giovanni.stelitano01@universitadipavia.it

**Abstract:** *Mycobacterium abscessus* is an emerging opportunistic pathogen that infects mainly the respiratory tract of individuals with pre-existing clinical pictures. In recent years, the incidence of infections of this microorganism has risen, in particular in patients with cystic fibrosis, leading to an exacerbation of their conditions. The actual therapeutic regimen has low efficacy and is extended for long periods since it is mainly based on a combination of repurposed drugs, generally from treatments of *Mycobacterium tuberculosis* infections. For this reason, it is necessary to develop new drugs or alternative strategies in order to improve the efficacy and shorten the time of treatments. This review aims to give an overview of drugs in the pre-clinical and clinical phases of evaluation against *M. abscessus* and the molecules that have been in development for the past five years in the early drug-discovery phase.

**Keywords:** *Mycobacterium abscessus*; inhibitors; drug target; therapy; enzymes





## 1. Introduction

*Mycobacterium abscessus* (*Mab*) is an environmental rapid-growing non-tuberculous mycobacterium (NTM) [1]. It was isolated in 1953 by Moore and Frerichs from a knee infection, marking the first recorded instance of its discovery [2]. In recent years, this peculiar bacterium has become extensively studied due to its similarities with *Mycobacterium tuberculosis* (*Mtb*) and as an opportunistic human pathogen.

In 2009, the *Mab* genome was sequenced for the first time, revealing a chromosome length of 5 Mb with a G+C content of 64% and approximately 49.920 predicted coding sequences. The bacterium also carries a mercury resistance plasmid identical to the one found in *Mycobacterium marinum*. Moreover, genes typical of *Actinobacteria* and *Pseudomonas* species have been identified, suggesting that they have been acquired by horizontal gene transfer (HGT) [3]. For example, the gene coding for the phospholipase C is more similar to the orthologous of *Streptomyces* species, *Chromobacterium violaceum*, and *Pseudomonas aeruginosa* than to the *Mtb* homologous, while the *mgtC* gene seems to be acquired from *Actinobacteria* by HGT, too [3]. These genes have proved to be important for the bacterium's survival in the human host [4]. The *Mab* genome contains an 81 kb full-length prophage, three prophage-like elements, and five inserted sequences [3].

It was hypothesized that this microorganism can be transmitted to humans through upstream water contamination [5]. Its capability to colonialize both the environment and humans is probably due to its ability to change its cell wall features. Indeed, the *Mab* morphotype can be rough R or smooth S, depending on the presence of glycopeptidolipids (GPLs) in the wall. GPLs facilitate pathogen–host interaction, biofilm formation, motility, and intracellular survival [4].

As an opportunistic pathogen, *Mab* may establish its infection in immunocompromised individuals or patients with chronic respiratory diseases, such as cystic fibrosis and bronchiectasis, especially in co-infection with other bacteria, such as *P. aeruginosa* [6].

The *Mab* mechanism of infection in humans is not fully understood, although several hypotheses have been proposed. The most plausible scenario suggests that *Mab* may initially colonize human cells, triggering recognition of the innate immune response and the subsequent recruitment of host immune cells, finally leading to the formation of granulomas. Although macrophages and neutrophils phagocytose bacterial cells, *Mab* can avoid intracellular destruction and continue the infection, leading to immune activation and recruitment of B and T lymphocytes. The irreversible change from S to R variant results in the formation of bacterial cords, which are able to evade phagocytosis and induce the auto-destruction of nearby tissues [7].

*Mab* infections may also be caused by the so-called *M. abscessus* complex that involves three organisms: *Mab* subspecies *abscessus*, *Mab* subspecies *massiliense,* and *Mycobacterium bolletii*. The complete 16S rRNA gene sequence is shared between *Mab* and the other two subspecies, but the further differentiation relies on differences in specific genes [8]; thus, drugs specific for the other NTM must be integrated to clear the infection.

*Mab*, likely *Mtb*, is resistant to many drugs through a plethora of different mechanisms. Its thick cell wall, for instance, is organized in a peculiar multi-layer complex that results in low permeability for many antibiotics [9]. This wall, together with the efflux pumps, represents the bacterial intrinsic resistance to antimicrobials (AMR). AMR may also be acquired from the environment, for example, by changes in the expression of genes and proteins following exposure to a drug (adaptive resistance) [4]. For example, *Mab* can modify a drug target to avoid recognition in the drug-binding site through N-acetyltransferase or phosphotransferase activities [10]. Finally, point mutations can alter the susceptibility to drugs thanks to the relative amino acid change that may interfere with the interactions between target and drug [4,11]. All these AMR mechanisms result in difficulties in the treatment of *Mab* infection.

## 2. Ongoing *M. abscessus* Therapeutic Regimen

*Mab* establishes its infection mainly in immunocompromised individuals or patients with chronic respiratory diseases, such as cystic fibrosis and bronchiectasis. In particular, the thick mucus of patients with cystic fibrosis (pwCF) facilitates *Mab* and other opportunistic pathogens to establish chronic infections and colonialize the respiratory tract [6].

The treatment of *Mab* infection, as well as other NTM, remains difficult because of its AMR mechanisms and persistence; thus, there is not a consolidated and standardized therapy yet. For example, the actual guideline against NTM pulmonary infections is based on a mix of oral antibiotics and macrolides, depending on the susceptibility of the isolates [12]. However, according to the guidelines published in 2015 by the US Cystic Fibrosis Foundation and European Cystic Fibrosis Society, the current recommendations to treat *Mab* infections in individuals with pulmonary diseases, such as pwCF, is based on oral macrolide treatment, usually azithromycin, combined with intravenous administration of amikacin (AMK) and one or more additional antimicrobials among the following: tigecycline, imipenem (IMI) and cefoxitin (CEF). Infection is monitored by sputum culture conversion, and the end of the treatment is followed by a continuation phase consisting of administration of oral or inhaled antibiotics for more than 12 months [13].

Drugs currently used against resistant *Mab* strains are mostly repurposed anti-TB antibiotics that target protein syntheses, such as azithromycins, AMK, and tigecyline, and cell wall biosynthesis, like IMI and CEF. Due to *Mab*'s capability to acquire resistance, developing new strategies to use in combination with the current antibiotic treatment is a priority. For example, the use of new or repurposed drugs and the combination with phage therapy or other antimicrobial agents, such as small antimicrobial peptides or nitric oxide (NO), could be integrated. NO administration by inhalation is safe and tolerable and showed a reduction of *Mab* load in sputum and improvement in lung function in initial clinic trials [6,14]. Thanks to these promising results, NO treatment is currently undergoing phase II clinical trials as a supplement to the main therapeutic regimen. Moreover, host-

directed therapy in combination with the classical antibiotic approach may be a valid help in *Mab* infection control [15].

To improve the effectiveness of therapies and shorten the administration time, the development of new drugs against Mab is required. This review aims to provide an overview of the compounds currently in clinical trials for the treatment of *Mab* infections, as well as the compounds in development in the early drug discovery phase in the past five years.

## 2.1. Preclinical Phase

Drugs that are classified as in the preclinical phase (Scheme 1) are used in treatments of other mycobacteria infections. They recently underwent evaluation for the treatment of Mab infections.

| Preclinical Phase | | |
|---|---|---|
| **Inhibitors of protein biosynthesis** | **Inhibitors of membrane biosynthesis** | |
| **Ribosome** Delpazolid Tedizolid | **MmpL3** PIPD1 | **HadA** Thioacetazone |
| **Inhibitors of β-lactamase** | **Inhibitors of ATP biosynthesis** | **Inhibitors of nucleic acids biosynthesis** |
| **Bla_MAB** Doripenem Avibactam Relebactam Vaborbactam | **F-ATP synthase** Bedaquiline | **RNA polymerase** Rifabutin |
| | **Unknown** Clofazimine | |
| **Phase I/II Clinical Trials** | **Phase IV Clinical Trials** | |
| **Inhibitor of cell wall biosynthesis** | **Inhibitor of protein biosynthesis** | |
| **Unknown** Clofazimine | **Ribosome** LZD | |

**Scheme 1.** The molecules under investigation against *M. abscessus* in the different stages of preclinical and clinical phases are reported in the scheme. The molecules are grouped according to their biological targets.

Delpazolid (LCB01-0371) is an oxazolidinone with a cyclic amidrazone, synthesized from linezolid (LIN) and firstly evaluated against multi-resistant *Staphylococcus aureus* (MRSA), showing good in vitro and in vivo bacteriostatic activities against Gram-positive bacteria [16]. Currently, in the phase 2 clinical trial for Gram-positive bacteria treatment and *Mtb*, its effectiveness is under evaluation against NTM infections [17]. In 2017, in vitro and in vivo studies of LCB01-0371 against wild-type *Mab* and clinical isolates showed that this compound inhibits bacterial growth in mouse lung infection models. Delpazolid was active against AMK, CEF, or clarithromycin (CLA)-resistant *Mab* with a $MIC_{50}$ of 1.2 μg/mL. Moreover, it showed an improved safety profile compared to LIN, with lower toxicity even maintaining a good pharmacokinetic profile [18]. LIN inhibits protein synthesis at an early stage by binding to the A-site pocket at the peptidyl transferase center (PTC) of the ribosome 50S subunit, thus preventing the binding of the initiator tRNA, as demonstrated by X-ray crystallization [19]. Isolation of delpazolid-resistant *Mab* mutants confirmed that this compound shares the same mechanism of action of LIN, although it is still unclear whether delpazolid binds PTC to the same site [18,20].

Indole-2-carboxiamide (IC) is a molecule with potent activity against *Mtb* [21]. Recently, through structure-activity relationship (SAR) analysis, some derivatives of this compound have been proven to possess a stronger effect against bacteria [21]. Two of them have also been tested against *Mab*, showing a MIC of 0.125 μg/mL [22].

Similarly, a different series of IC derivatives active against *Mab* has been tested on acute toxicity mouse models, demonstrating that these molecules are effective in vivo, leading to a reduction of bacterial load in the lungs and good oral bioavailability [23]. It was proven that IC and derivatives disrupt cell wall biosynthesis by inhibiting MmpL3, a transmembrane protein involved in the transport of trehalose monomycolate onto the cell surface. Strains harboring single-point mutations in the MmpL3 gene showed high resistance phenotypes to IC derivatives [22].

PIPD1 is a novel piperidinol derivative that has been identified through the screening of a library of 177 molecules active against *Mab*. PIPD1 showed bactericidal activity against *Mab* clinical isolates with a MIC of 0.125 μg/mL, with the same mechanism of action of IC and derivatives. The PIPD1 effect was also evaluated in zebrafish infection models, reducing the bacterial burden and decreasing larval mortality [24].

Thioacetazone (TAC) is a bacteriostatic agent used for *Mab* treatment in combination with isoniazid (INH) and streptomycin (STR) in Africa and South America. The TAC mechanism of action was identified in *Mtb*. It is a prodrug activated through oxidation of its thiocarbonyl moiety by the monooxygenase EthA. Once activated, it binds to HadA, a component of the HadABC dehydratase complex involved in fatty acid biosynthesis, resulting in the inhibition of mycolic acid biosynthesis [25]. *Mab* is intrinsically resistant to this compound, but some TAC derivatives have shown promising inhibitory effects on *Mab* growth, with MIC values ranging between 1.6 and 6.2 μg/mL [26]. This data suggests that structural modification of the TAC scaffold could provide new effective drugs not only against *Mtb* but also against NTM.

Clofazimine (CFZ) is a lipophilic drug used for the treatment of *Mycobacterium leprae*. It acts as an artificial electron acceptor, causing respiratory chain malfunction and the disruption of ion transport across the cell wall. This double effect leads to a reduction of ATP availability and an increase in the reactive oxygen species inside the bacterial cell [27]. The combined treatment of CFZ with other common *Mab* antibiotics resulted in improved outcomes, with an 81% treatment response rate based on symptoms, but further evaluations of its efficacy are required [26]. Moreover, CFZ showed significant synergistic activity with AMK in vitro in 82% of cases against *Mab* clinical isolates [28], suggesting its possible use in combination with current anti-*Mab* drugs.

Bedaquiline (BDQ) is a repurposed anti-tubercular drug that targets the bacterial respiratory chain, inhibiting the mycobacterial $F_0F_1$ ATP synthase, also known as F-ATP synthase. It has been proved in vitro and in zebrafish infection models that BDQ possesses a broad-spectrum bacteriostatic activity against NTM. Moreover, it showed MIC values ranging from 0.031 to 0.125 μg/mL against different *Mab* clinical isolates [29]. The BDQ mechanism of action in NTM was confirmed by isolating low-resistant *Mab* strains and inserting two single-point mutations in the AtpE gene. The derived mutants present the amino acid substitutions D29V and A64P, showing high resistance to BDQ [29]. Its efficacy against *Mab* has been largely proved by infecting different mouse models with one *Mab* clinical isolate from pwCF and then treating it with the drug [30]. In this study, three mouse models were used: the severe combined immunodeficiency model (SCID), the granulocyte–macrophage colony-stimulating factor (GM-CSF) knockout model, and the nude mice model, all showing similar symptoms to human pulmonary *Mab* infection. In all models, the administration of BDQ for 5 days decreases the bacterial load in the spleen and liver. Moreover, after 8 days of treatment, the mycobacterial load also decreased in the lungs [30]. The use of BDQ in combination with CFZ decreased the mycobacterial burden in the spleen, liver, and lungs only after five days, showing a synergistic effect of the two compounds [30]. Finally, since efflux pumps require the proton motive force and adenosine triphosphate (ATP) to export molecules outside the cell, BDQ indirectly impairs this mechanism of drug extrusion, providing a powerful integration into the actual therapeutic regimen [29].

Rifamycins, such as rifampin, are antitubercular drugs ineffective against *Mab* since they are inactivated by ribosylation of their scaffold at the C23 hydroxyl position by the

enzyme ADP-ribosyltransferase (Arr$_{Mab}$). However, the screening of libraries of approved antitubercular drugs led to the discovery of rifabutin (RIF), a compound of this family that displays bactericidal activity against *Mab* [31]. It was demonstrated in a macrophage infection model that administration of RIF decreases intra- and extracellular cording of the *Mab* R morphotype. The same result was observed in *Mab*-infected zebrafish following RIF treatment, together with a reduced abscess formation [32].

*Mab* exhibits intrinsic resistance to β-lactams except for CEF and IMI, which are currently used in treatments of *Mab* infections. This dual β-lactams combination showed a synergist effect against all of the considered *Mab* clinical isolates, while the combination of Doripenem and IMI resulted in a synergetic effect against most strains. This outcome is probably due to the different targets hit by these two β-lactams, possibly non-redundant enzymes involved in the biosynthesis of the cell wall, consequentially leading to a synergistic effect [33]. Different β-lactams combinations were also tested in vivo in the *Mab*-infected murine model, according to the synergy showed in vitro. After four weeks of treatment, the average lung CFU decreased by 6 log$_{10}$ compared to the untreated mice [34]. These results confirmed that the current *Mab* treatment could benefit from the administration of combined β-lactams. However, the insurgence of resistant strains remains a challenge, limiting the use of such compounds. Resistance to β-lactams in *Mab* is mediated by a broad-spectrum Ambler-Class A β-lactamase (Bla$_{MAB}$), which hydrolyses common β-lactamase inhibitors such as clavulanic acid, tazobactam, and sulbactam [35]. Therefore, many studies focused on finding new β-lactamase inhibitors to use in combination with β-lactams. For example, avibactam is a non β-lactam β-lactamase inhibitor used in combination with ceftazidime for the treatment of Gram-negative bacterial infections [36]. Avibactam was tested in combination with several representatives of the three main classes of β-lactams (penams, cephalosporins, and carbapenems) against a set of *Mab* clinical isolates. In the presence of this molecule, even compounds usually ineffective against *Mab* inhibited its growth [37]. Moreover, the combination of amoxicillin and avibactam was evaluated ex vivo in the intracellular infection model and in vivo in the zebrafish model. In both cases, the co-administration of avibactam increased the activity of amoxicillin, resulting in an increased survival rate and a decrease in abscess formation in infected zebrafish larvae [37]. Another study evaluated the effect of Avibactam in combination with piperacillin in vitro against *Mab* and in vivo in a *Galleria mellonella* infection model. The combination resulted in a lower bacterial burden compared to the larvae treated with piperacillin only. Moreover, the in vitro MIC was 16–32 folds lower when the two compounds were used in combination [38].

Relebactam and vaborbactam are the other two non β-lactam β-lactamase inhibitors recently approved for the treatment of bacterial infection in combination with IMI and meropenem. These compounds were recently evaluated against *Mab* clinical isolates in combination with a wide range of carbapenems and cephalosporins, resulting in a significant decrease of MIC$_{50}$ and MIC$_{90}$ by one or two dilutions, respectively [39]. These results confirm that the combination of non-β-lactam β-lactamase inhibitors with β-lactams is effective for the treatment of *Mab* infections with positive outcomes and to expand the range of effective β-lactams against these bacteria.

Tedizolid (TDZ) is a next-generation oxazolidinone antibiotic approved by the Food and Drug Administration in 2014 for the treatment of soft tissue and skin infections caused by some Gram-positive bacteria, such as *Streptococcus* spp. and MRSA. The efficacy of TDZ was evaluated in vitro against a pool of *Mab* clinical isolates both alone and in combination with other antimicrobials, such as doxycycline, AMK, and CLA. TDZ showed bacteriostatic activity against the considered strains without instances of antagonism with other antimicrobials [40]. Another study confirmed the efficacy of TDZ in vitro with MIC values ranging between 0.25 and 8 mg/L depending on the strain and in an intracellular macrophage infection model. Moreover, at a concentration of 2x MIC, TDZ reduces bacterial load by a factor of 500 compared to the untreated condition [41]. This molecule was also evaluated on twelve patients who met the ATS/IDSA criteria for NTM infections. The treatment was

based on a multi-drug therapy, TDZ included, showing benefit from this integration with improvement in symptoms and the clinical outcome [42].

*2.2. Phase II Clinical Trial*

Despite the lack of any new drug in phase I clinical trials, some studies are focusing on different delivery methods, such as the Liposomal AMK for Inhalation (LAI), which is in phase II of clinical trials. This new formulation is composed of dipalmitoylphosphatidilcholine (DPPC) and cholesterol-containing encapsulated AMK to be delivered through aerosol. AMK is a semi-synthetic aminoglycoside antibiotic derived from kanamycin and commonly used in *Mab* treatment. It binds the 30S ribosome subunit, inhibiting protein biosynthesis [43].

LAI preparation is capable of lowering drug toxicity compared to systemic AMK administration. Moreover, a higher concentration of AMK is maintained at the site of infection, allowing a lower dosage, thus reducing the potential side effects. LAI was evaluated in vitro against *Mab* in a macrophage infection model, resulting in better activity compared to free AMK at a dose of 10 mg/L [44]. During a randomized phase II clinical trial, three *Mab*-infected patients with persistently positive NTM cultures, even after American Thoracic Society/Infectious Diseases Society of America (ATS-IDSA) guidelines-based treatment, were treated with LAI preparation achieving culture conversion [45]. The phase II trial is currently recruiting to further evaluate this formulation in order to confirm its efficacy against *Mab* and to assess its safety for infected patients [17].

*2.3. Phase IV Clinical Trial*

LZD is an oxazolidinone currently undergoing phase IV clinical trials for the treatment of NTM infections. Similarly to the two derivatives, TDZ and delpazolid, LZD inhibits protein synthesis by binding the 23S ribosome subunit to the binding site of the initiator tRNA, thus blocking the translation process at its beginning [19]. In 2018, LZD was used to treat a multi-drug-resistant *Mab* pulmonary infection case, leading to the successful recovery of the patient [46]. Despite its activity against NTM and Gram-positive bacteria, including MRSA, vancomycin-resistant enterococci (VRE), and penicillin-resistant pneumococci, prolonged administration of LZD is advised against due to its toxicity. Indeed, LZD causes long-term side effects, such as reversible myelosuppression, potentially irreversible optic neuropathy, and peripheral neuropathy [47].

## 3. Early Drug Discovery Phase

The molecules that have been under development for the last five years in the early drug-discovery phase against *Mab* are discussed below and shown in Scheme 2. This section is organized according to the metabolic pathway targeted.

*3.1. Inhibitors of Membrane Biosynthesis*

One of the most interesting strategies to kill mycobacteria is hitting the cell wall biosynthesis. This is possible by exploiting different strategies and targets, such as lipids transporters, penicillin-binding proteins, or mycolic acid metabolism.

| Early Drug Discovery Phase | | |
|---|---|---|
| **Inhibitors of membrane biosynthesis** | | |
| **MmpL3**<br>IC5<br>IC25<br>HC2091<br>HC2134<br>CRS400155<br>CRS400226<br>HC2099<br>E11<br>EJMCh-6<br>SQ109 | **Ag85**<br>iBpPPOX<br>**Ldt_{MAB1,2,4,5(H6)}**<br>Ceftaroline<br>Imipenem | **Bla_{MAB}**<br>Durlobactam<br><br>**Rv3802c**<br>Phenyl-urea based 4e compound |
| | **InhA**<br>NITD-916 | **MmpL4**<br>CyC25<br>CyC26 |
| | **DprE1**<br>OPC-167832 | |
| **Inhibitors of efflux pumps**<br><br>**Efflux pumps proteins**<br>NUNL02<br>CCCP | **Inhibitors of ATP biosynthesis**<br><br>**F-ATP synthase**<br>WX-081<br>TBAJ-876<br>GaMF1 | **Inhibitors of NAD(H) metabolism**<br><br>*Mab* **NadD**<br>N2-1<br>N2-11 |
| **Inhibitors of nucleic acids biosynthesis** | | **Inhibitors of coenzyme A biosynthesis** |
| **DNA gyrase**<br>TTP8<br>SPR719<br>884<br>884-TMF<br>EC/11716 | **Polimerases**<br>CGM<br>MMV688845<br>Rifamycin O | **PanD**<br>POA<br><br>**CoaD**<br>Fragment 5<br>Fragment 20 |
| **Inhibitors of protein biosynthesis** | | |
| **Ribosome**<br>Aminomethylcycline omadacycline<br>Halogenated tetracycline eravacycline<br>Glycylcycline tigecycline | **TrmD**<br>AW5<br>AW6<br>AW7<br>**ClpC1**<br>RUFI<br>Compound 12 | **LeuRS tRNA**<br>MRX-6038<br>EC/11770 |
| **Inhibitors of virulence factors** | | **Unknown targets** |
| *Mab*-**SaS**<br>5-phenylfuran-2-carboxylic acid-derived molecules (Compound 1) | **Iron mimic**<br>GaPP<br>GaMP<br>Ga(NO_3)_3 | 10-DEBC |

**Scheme 2.** The scheme shows the molecules in early phase drug development against *M. abscessus* grouped according to their biological targets. Each color is associated with a biosynthetic pathway.

### 3.1.1. Inhibitors of Transporter Proteins

The inhibition of the mycobacterial membrane protein Large 3 (MmpL3), a lipid transporter involved in the translocation of the trehalose mono-mycolate (TMM) towards the cell wall, is by now an attractive and validated strategy to kill mycobacteria [23,48].

The indole-2-carboxamides (ICs) are molecules that interfere with the mycolic acids' transport across the mycobacterial inner membrane by targeting MmpL3 [21,49,50]. Recently, they have been tested against *Mab*, too. Two compounds from this class, the IC5 and IC25 shown in Table 1, exhibited MIC values of 0.25 μg/mL and 0.063 μg/mL, respectively, against *Mab* ATCC19977 with bactericidal effect. The target of these compounds was confirmed in *Mab* by genetic studies over a mutant strain with a missense mutation on the mmpL3 gene, which showed a high increase of both MICs of ~100 times [23]. These two IC molecules have been well tolerated in vivo, as shown in acute toxicity mouse models at different dosages and for different days of administration without negative side effects. The pharmacokinetic analysis of both molecules showed interesting results in terms of maximum drug levels, half-life of the compounds, plasma clearance, and volume of distribution. Moreover, control-infected mice showed a log_{10} CFU of 6 for lung, spleen, and liver

against the $\log_{10}$ CFU of 3–5 in the same organs of treated mice for both compounds. These results confirm the possibility of repurposing the ICs against *Mab* infections.

**Table 1.** Structures of the above-discussed inhibitors of the transporter protein.

| Code | Structure | Code | Structure |
|------|-----------|------|-----------|
| IC5 |  | IC25 |  |
| HC2091 |  | CRS400226 |  |
| E11 |  | EJMCh-6 |  |
| SQ109 |  | iBpPPOX |  |

Other derivatives of the same scaffold that had a preliminary evaluation are the *N*-[2-(4-chlorophenyl)ethyl]-4-thiophen-2-yloxane-4-carboxamide (HC2091, Table 1) exhibiting a $MIC_{50}$ of 6.25 μM and *N*-(2-methoxy-5-nitrophenyl)-1-oxo-4-phenylisochromene-3-carboxamide (HC2134) that exhibited a $MIC_{50}$ of 12.5 μM [51]. Moreover, they have been tested only in vitro against *Mab*, and further evaluation is necessary to prove that they are also active ex vivo and in vivo in animal models.

The benzothiazole amide derivatives share a similar scaffold and mechanism of action with ICs [52]. The best substituents in benzothiazole-based compounds have been identified in adamantly and cyclohexyl derivatives groups, which effectively inhibit the rapid-growing NTM class, *Mab* included. These compounds showed great MICs over *Mab* ATCC19977. From the adamantyl benzothiazole amide, named CRS400226, two other compounds have been developed, further improving the inhibition power of this scaffold. The MICs of these compounds ranged between 0.03 and 0.06 μg/mL for the CRS400359 and CRS400393 and were 0.25 μg/mL for CRS400226 and 0.5 μg/mL for CRS400153 [53]. In particular, CRS400153 and CRS400226 have been exploited for the identification of the mechanism of action monitoring the cell wall biosynthesis. At a concentration of 2X and 10X MIC, they inhibited MmpL3 transport since the mycolic acid was not observed in the mycobacterial cell envelope. This inhibition was proved to be concentration-dependent. Moreover, *Mab* mutant strains resistant to the benzothiazole amides showed mutations in the MmpL3 transporter, confirming that MmpL3 is the target [53]. In vivo, analysis of CRS400226 in a chronic *Mab* infection mouse model confirmed that this compound is efficient and tolerable since no side effects have been shown after the infection was cleared [53]. CRS400226 structure is shown in Table 1.

Another derivative of benzothiazole amide under evaluation is the 2-[(6-chloro-1H-benzimidazol-2-yl)sulfanyl]-*N,N*-di(propan-2-yl)acetamide HC2099, which showed a MIC of 25 μM against *Mab* [51]. It has been evaluated only in vitro against *Mab*; hence, further studies are necessary to prove its efficacy in animals and humans.

A different scaffold capable of targeting MmpL3 is well-represented in the acetamide E11 compound (Table 1). It has been well-characterized in *Mtb* and subsequently evaluated

against *Mab* since their similarities, showing a MIC of 12 μM against the ATCC19977 strain and 25 μM against the *Mab* subsp. *abscessus* Bamboo clinical isolate [54]. Additionally, the preliminary studies of E11 are limited to in vitro and in vivo evaluation against the bacteria, so further evaluations are necessary before moving into clinical.

Among the compounds that impair the MmpL3 transporter, the 2-(2-phenalkyl)-1Hbenzo[d]imidazole scaffold has been identified in *Mtb* and subsequently evaluated over *Mab*. For example, the 2-(2-cyclohexylethyl)-5,6-dimethyl1H-benzo[d]imidazole (EJMCh-6, Table 1) inhibits the trehalose monomycolate (TMM) transport decreasing the arabinogalactan mycolylation [55]. Indeed, the in vitro evaluation of EJMCh-6 showed *Mab* growth inhibition with a MIC of 0.125 μg/mL and a bacteriostatic effect. Moreover, it showed MIC values of 0.031 μg/mL against the non-CF R clinical isolates CIP104536 belonging to the *Mab* complex and 1 μg/mL for the same CF S variant. Nevertheless, similar MIC values resulted from both R and S strains, proving its high efficacy against *Mab* [55]. EJMCh-6 also showed low toxicity against human THP-1 macrophages; hence, it has been evaluated with a dosage of 3 μg/mL over S *Mab* infecting macrophages. Results confirmed a reduced mycobacterial load at 3 dpi. In zebrafish, the administration of 0.75 μg/mL of EJMCh-6 resulted in 80% of alive embryos compared with only 40% surviving untreated animals [55]. The presence of single nucleotide polymorphism (SNP) in the mmpL3 gene in EJMCh-6-resistant strains suggests that this transporter is the target. This is further confirmed by the overexpression of mutated mmpL3 strains, which showed increased MIC values [55].

Ethylenediamines are a known class of inhibitors of *Mtb* mycobacterial membrane biosynthesis. This class of compounds has been recently repurposed for use against *Mab*. The *N*-geranyl-*N'*-(2-adamantyl)ethane-1,2-diamine, also known as SQ109 (Table 1), for example, exhibited a MIC 0.5 μM against *Mtb*, whereas higher values of 22 μM and 44 μM against R *Mab* and S *Mab,* respectively [56]. On the contrary, two analogs characterized by the n-butyl and the benzyl groups showed better MICs of 6 μM each against both morphotypes. A resistant *Mab* mutant strain in the mmpL3 gene was used to confirm that the related protein is the compound's main target. This strain, which showed resistance to all the major inhibitors of MmpL3, was sensitive to SQ109, which showed a MIC comparable to wild-type *Mab*. This finding suggests that MmpL3 may be a secondary target for this class of inhibitors in *Mab* [56].

The oxadiazolone scaffold has been found to be effective against *Mab*, too. In particular, the iBpPPOX derivative (Table 1) showed a MIC value of 33.0 μM for the S morphotype and about 50 μM for the R morphotype. It seems clear that the effectiveness of iBpPPOX against *Mab* depends on the position of the phenoxy group compared to other derivatives of the same class [57]. The intracellular effect of this molecule has been evaluated against the S morphotype in the Raw264.7 murine macrophage infection model, showing that every concentration tested resulted in the same bacterial killing effect of about 58%. This may be due to the stress dictated over the metabolism, which led to bacterial death. Moreover, the intracellular concentration of 30 μM gave the same result, confirming the extracellular MIC [57]. The target of iBpPPOX is the Ag85 protein, which carries out its activity in cell wall biosynthesis. This has been proven by the overexpression of Ag85, which led to an increased MIC, while its genetic inactivation showed no differences compared to the wild type after the iBpPPOX administration [57].

### 3.1.2. β-Lactamases Inhibitors in Combination with β-Lactams

As briefly discussed in the preclinical phase, the β-lactams disrupt the peptidoglycan biosynthesis, and resistance to these drugs is particularly important in mycobacteria, *Mab* included [58]. For this reason, different combinations of β-lactams in combination with β-lactamases inhibitors are in development [59].

Molecules belonging to the diazabicyclooctanes (DBOs) class are known inhibitors of β-lactamases enzymes. A brand-new DBO is the durlobactam (DUR), which has been improved compared to the previous DBO ring-featured molecules, and it is currently in clinical trials against other microorganisms in combination with β-lactams [60]. DUR

has recently been evaluated against *Mab*. This molecule targets the Bla$_{MAB}$ β-lactamase enzyme-binding tightly, as demonstrated by its Ki of $4 \times 10^{-3}$ µM, about 75-fold lower than avibactam and relebactam [60]. The MIC of DUR was evaluated against a hundred-one *Mab* clinical isolates, resulting against all strains in the range of 2–8 µg/mL. In combination with amoxicillin or IMI (ratio 1:1), it decreased at 4 µg/mL [60]. Moreover, the triple combination of DUR, amoxicillin and IMI exhibited a MIC between 0.06 and 2 µg/mL, while the combination of DUR, amoxicillin, and cefuroxime showed a MIC between 0.06 and 1 µg/mL [60]. These results confirm that this novel β-lactamase inhibitor could successfully be integrated into β-lactams treatments to prevent their hydrolysis.

Additionally, not all the combinations of β-lactam and β-lactamase inhibitors are effective. For example, ceftaroline, a β-lactam already in clinical use, has been evaluated against *Mab* ATCC19977, showing a MIC of 16 µg/mL. A mild synergistic effect in combination with the β-lactamases inhibitors relebactam decreased the MIC of 1–2 dilutions while avibactam decreased the MIC of 4 or more dilutions [61]. In contrast, the combination of IMI, another β-lactam used in the clinic, with both the previous compounds maintained the MIC value unaltered at 2 µg/mL. To better understand these differences in synergy, these β-lactamases inhibitors have been evaluated against the Bla$_{MAB}$ enzyme from a kinetic perspective, confirming that avibactam inhibits the enzyme better than relebactam. Indeed, the avibactam K$_d$ was lower than relebactam, indicating a stronger binding even if they show similar K$_{off}$ values [61]. Moreover, the two β-lactams have been evaluated against different Mab enzymes that are part of the L-D-transpeptidases family (Ldts), showing that the binding affinities (K$_i$) of IMI for Ldt$_{MAB1}$, Ldt$_{MAB2}$, and Ldt$_{MAB4}$ are higher than ceftaroline, while both IMI and ceftaroline have a similar affinity for Ldt$_{MAB5(H6)}$. These data further confirm that the combination of ceftaroline and IMI with β-lactamases inhibitors may be an effective strategy against *Mab* [61].

### 3.1.3. Other Inhibitors of the Cell Wall Metabolism

Rv3802c belongs to a gene cluster that modulates mycolic acid biosynthesis and transport. Recent studies have confirmed that this enzyme hydrolyses the phosphatidylinositol, regulating the membrane lipid content [62]. Considering the importance of his role in cells, Rv3802c has been taken into consideration as a putative drug target to kill mycobacteria. Urea-based molecules have been identified as strong hydrolase inhibitors [63]; hence, heterocyclic urea derivatives have been studied against *Mtb* hydrolases. Considering the results of these compounds, a selection of derivatives has been evaluated against *Mab* ATCC19977, too. Only the 4e compound, shown in Table 2, exhibited a moderate MIC of 25 µM, while the toxicity analysis performed over murine cells showed a higher MIC. Despite these preliminary results, this study is a valuable starting point for the development of effective *Mab* hydrolase inhibitors [62].

**Table 2.** Structures of a selection of the above-discussed inhibitors of enzymes involved in cell wall metabolism.

| Code | Structure | Code | Structure |
|---|---|---|---|
| 4e |  | NITD-916 |  |
| OPC-167832 |  | CysC26 |  |

Another important target involved in cell wall biosynthesis is the enoyl-ACP reductase (InhA), which is involved in mycolic acid biosynthesis [64]. INH effectively targets *Mtb* InhA upon activation by the catalase-peroxidase KatG. Recently, this target has also been studied in *Mab*, using NITD-916 (Table 2), a 4-hydroxy-2-pyridone that bypasses the activation process of the KatG and directly inhibits InhA. The MIC against *Mab* was variable, depending on the medium used, and ranged from 0.195 µg/mL to 6.25 µg/mL [64]. To note, the killing effect of NITD-916 was not concentration-dependent, while the compound seemed to act as bacteriostatic [64]. NITD-916 was effective ex vivo in the human THP-1 macrophage infection model at a concentration of 12.5 µg/mL, which was below the observed cytotoxic effect. Positive results were also found in airway organoids derived from the lungs of healthy people and pwCF, effectively reducing *Mab* load in both conditions at a concentration of 15.6 µg/mL. Finally, to prove that NITD-916 is not activated by KatG, its MIC was evaluated in the presence of a *Mab* strain overexpressing this enzyme. The unaltered MIC against this strain confirmed the direct inhibitory effect of this molecule [64].

One of the most important enzymes involved in the biosynthesis of the mycobacterial cell wall is the decaprenylphosphoryl-b-D-ribose oxidase DprE1, which has been widely characterized in *Mtb* [65,66]. It catalyzes the formation of a precursor of the arabinogalactan, an essential player in the mycobacterial membrane. Preliminary repurposing attempts of the best molecules developed against *Mtb* DprE1 have been obtained against the homologous enzyme of *Mab*. The inhibitor dihydrocarbostyril OPC-167832 (Table 2), for instance, has been evaluated against *Mab* ATCC19977, showing a MIC of 5.3 µM, hence displaying a good activity against this strain [67]. Other MICs against *Mab* subsp. *Bolletii* CCUG 50184T, *Mab* subsp. *Massiliense* CCUG 48898T, and a panel of clinical isolates ranged between 5 and 15 µM, suggesting that the repurposing of OPC-167832 was not effective against *Mab* since the MIC in *Mtb* was assessed in nanomolar concentration. Moreover, in vivo analysis of OPC-167832 in different mouse models showed a non-statistically significant decrease of bacterial burden in lung and plasma, confirming the previous data [67]. However, the understanding of the molecular interactions between OPC-167832 and its target in *Mab* will probably allow us to improve its effect or design new inhibitors.

Another appealing target to fight NTM may be the metabolism of GPLs and their translocation on the R strain surface. Indeed, the presence of these glycopeptidolipids in the S morphotype of *Mab* sensibly decreases its susceptibility to many drugs [68]. For example, the multi-target inhibitors derived from cyclipostins and cyclophostins (CyC) showed higher MIC values against the S *Mab* morphotype, which has a cell wall rich in GPLs. Indeed, CyC17 and CyC18b against S *Mab* showed MICs of 6.4 and 6 µg/mL, respectively, while against R *Mab*, their MICs were 0.18 and 4.9 µg/mL [68]. Among the CyC derivatives that are characterized by a long lipophilic chain and share the phosphate ester scaffold, two features that help the crossing through the membranes, CyC25 and CyC26, showed $MIC_{50}$ of 13.9 µM and 6.9 µM against R *Mab*, respectively. The same compounds have $MIC_{50}$ close to 100 µM against the S morphotype, which means around 10 times higher [69]. The importance of GPLs in *Mab* resistance was confirmed by inactivation of *mmpL4b* in S *Mab* since the MmpL4b protein is involved in the translocation of GPLs in the cell wall. The same complemented strain regained resistance to CyC25 and CyC26 (Table 2), exhibiting high MIC values compared to the Δ*mmpL4b* S *Mab* [69]. This data suggests that the absence or reduction of GPLs may induce some changes in the cell-wall fluidity and permeability, possibly favoring the penetration of these inhibitors. Finally, the low toxicity of these molecules allows their use in combination with the well-established antibiotic regimen [69].

### 3.2. Inhibitors of Efflux Pumps

Efflux pumps are involved in the molecular mechanisms of AMR in bacteria, as previously mentioned in this paper. Because of their role, they are recently considered an attractive target for drug development. Inhibitors of the efflux pumps (EPIs) work synergistically with drugs, improving the accumulation of toxic molecules inside the bacterial cell. Along with the already-known *Mtb* EPIs such as verapamil, phenothiazines class

of molecules, protonophores, valinomycin, and plant-derived EPIs [70], EPIs specifically designed against this pathogen are under development.

Usnic acid, a compound of lichen derivation, showed antimicrobial activity against *Mtb* and different NTM, *Mab* included. The increase of ethidium bromide inside *Mab* cells and the decrease of the MIC of AMI and CLA of 4-fold in the presence of this compound suggest its inhibitory effect on efflux pumps [70].

Recently, the effect of NUNL02 (Table 3), a derivative of tetrahydropyridines and a known inhibitor of the *E. coli* AcrB efflux pump [71], was assessed against *Mab* ATCC19977 and three clinical isolates: *Mab* subsp. *abscessus* AT 07, *Mab* subsp. *bolletii* AT 46, and *Mab* subsp. *bolletii* AT 52, showing MIC values ranging from 50 to 200 µg/mL [72]. In this study, the MIC of AMI, ciprofloxacin (CIP), and CLA was evaluated in the presence and absence of NUNL02, and the known EPI verapamil was used as control. The effect of both compounds was assessed at the subinhibitory concentration of $\frac{1}{2}$ MIC, which does not affect the viability of bacteria [72]. Under these conditions, the presence of verapamil reduced the MIC of antimicrobials by 8-fold, while NUNL02 was 16-fold. Additionally, the ethidium bromide accumulation test further suggests that NUNL02 targets the efflux pumps, possibly confirming its mechanism of action in *Mab*. Additionally, this study lacks any in vivo evaluation to validate the potentiality of NUNL02 as an addition to therapies.

**Table 3.** Structures of NUNL02 and carbonyl cyanide 3-chlorophenylhydrazone CCCP, two inhibitors of *Mab* efflux pumps.

| Code | Structure | Code | Structure |
|---|---|---|---|
| NUNL02 |  | CCCP |  |

Carbonyl cyanide 3-chlorophenylhydrazone, CCCP (Table 3), is another EPI under investigation against *Mab*. The accumulation assay showed that CCCP lowers the MIC of CLA in different *Mab* CLA-resistant clinical isolates by 52.6%, while it did not show any effect against sensitive *Mab* strains [73]. The only strains resistant to CLA in the presence of CCCP presented the mutation *rrl* 2270/2271, which confers intrinsic resistance to CLA [73]. Recently, the effect of CCCP was evaluated against 47 *Mab* strains without the presence of any antibiotics to evaluate its toxicity on bacteria. Data showed MICs lower than 7.5 µg/mL against all strains [74]. The CCCP effect was also evaluated at a concentration of 6 µg/mL in a macrophage infection model. This experiment showed a concentration-dependent inhibition of the intracellular *Mab* growth of 84.8 ± 8.8% and of the *Mab* subsp. *massiliense* growth of 72.5 ± 13.7%. Its cytotoxicity was assessed on THP-1 and U937 cells, demonstrating that the compound is safe at concentrations ≤ 6.25 µg/mL, with over 97% cell viability [74]. Additionally, CCCP lacks the in vivo evaluation before progressing to clinical trial.

### 3.3. Inhibitors of ATP Biosynthesis

Mycobacteria produce ATP via oxidative phosphorylation through the F-ATP synthase that possesses peculiar features compared to the orthologous of other bacteria. Indeed, the subunit α contains a C-terminal elongated domain that regulates the speed of ATP synthesis. This domain is essential during the bacterial dormancy state [75]. The mycobacterial δ subunit has an inserted domain that interacts with α and β subunits, differently to other bacteria, chloroplast, and mitochondria where the δ orthologous interacts only with the α subunit [76]. Finally, the rotating central γ subunits contain an extra loop of 12–14 polar amino acids that are not present in any other prokaryotic or eukaryotic homologs and

are located in the proximity of 4 polar residues of the c ring. This loop is a regulatory element of the F-ATP synthase involved in the coupling of proton translocation with ATP synthesis [77]. These features are unique, attractive elements for drug development. New inhibitors of *Mab* ATP synthase are being developed to fight resistant strains to BDQ. These molecules may be BDQ analogs or derivatives, or they may exploit different scaffolds.

WX-081 (Table 4), known as sudapyridine, is a structural analog of BDQ, possibly with the same mechanism of action [78]. It was evaluated in vitro against 50 strains of NTM mycobacteria, showing a comparable MIC to BDQ for most of them [79]. In fact, the observed MIC values were mostly below 0.25 μg/mL and in no case above 2 μg/mL. Against *Mab*, the MIC determined in vitro was 0.25 μg/mL. WX-081 resulted in a similar cytotoxicity to BDQ but with fewer cardiac side effects, making this compound an interesting substitute for the classic drug [79]. However, this preliminary study lacks ex vivo and in vivo evaluation to demonstrate the real efficacy and limitations of WX-081.

**Table 4.** Structures of *Mab* F-ATP synthase inhibitors WX-081 and GaMF1.

| Code | Structure | Code | Structure |
|------|-----------|------|-----------|
| WX-081 |  | GaMF1 |  |

Another analog of BDQ is TBAJ-876, a diarylquinoline compound that targets the F-ATP synthase with the same mechanism of action [80]. It showed a bacteriostatic effect against NTM, including various strains of *Mab*, with a 2-fold lower MIC compared to BDQ [81]. In a murine infection model, it showed a similar effect to BDQ at a dose of 10 mg/Kg. Moreover, it did not exhibit antagonism with commonly used anti-*Mab* drugs in checkerboard assay [81], suggesting that TBAJ-876 could be a valuable integration into the actual therapeutic regimen.

GaMF1 (Table 4) is a known bactericidal compound selective for *Mtb* [82]. Its mechanism of action was hypnotized by in silico docking studies. It has been demonstrated that GaMF1 inhibits the rotation of the ATP synthase γ subunit by binding its extra loop. This interaction disrupts the function of the c-ring turbine, interrupts proton flux, and ultimately hinders ATP synthesis [82]. Recently, GaMF1 was evaluated against *Mab* subsp. *abscessus* ATCC19977, showing a $MIC_{50}$ of 33 μM in Middlebrook 7H9 broth and 13 μM in cation-adjusted Mueller–Hinton (CAMH) medium [83]. Similar $MIC_{50}$ values were observed for the clinical isolate *Mab* Bamboo. Moreover, the bactericidal effect of GaMF1 was observed 10-fold the $MIC_{50}$ in both media [83]. GamF1 was analyzed against the enzyme by an intracellular ATP synthesis assay and by inside-out membrane vesicles (IMVs), showing an $IC_{50}$ of $10 \pm 0.9$ μM against the ATP synthesis and of $13 \pm 2.5$ μM against the NADH-driven ATP synthesis [83]. Finally, the combination of GaMF1 with CFZ, RIF, or AMI revealed an increased susceptibility of *Mab* to these drugs.

### 3.4. Inhibitors of NAD(H) Metabolism

NAD(H) is an essential redox cofactor for many enzymes, such as the ones involved in glycolysis, and it is an electron donor in the respiratory chain. Depletion of NAD(H) has been shown to affect both these processes, leading to the killing of mycobacteria [84–86]. Consequently, NAD biogenesis has been identified as a valuable target to impair mycobacterial growth [87]. In addition, NAD biogenesis inhibition could synergistically impair ATP synthesis [84].

NadD is a nicotinic acid mononucleotide (NaMN) adenylyltransferase implicated in NAD biosynthesis and the most divergent enzyme from its human counterpart [86]. In the

literature, different series of molecules selectively hit *Mtb* NadD are reported. One of this series, named N2, comprises a benzimidazolium core that interacts with the catalytic amino acids Asp109 and Glu160, perturbing the NadD dimer quaternary structure and leading to the inhibition of the enzymatic activity. Although the series was evaluated in vitro against the *Mtb* homologous enzyme, it was also tested against *Mab*. The most effective compounds (Table 5) exhibited MIC values in the micromolar range. Specifically, N2-1 showed a high MIC of 61.4 μM while N2-11 had a MIC of 26 μM [87].

**Table 5.** Structures of *Mab* NadD enzyme inhibitors N2-1 and N2-11.

| Code | Structure | Code | Structure |
|---|---|---|---|
| N2-1 |  | N2-11 |  |

## 3.5. Inhibitors of Nucleic Acids Biosynthesis

Drugs may inhibit different enzymes involved in nucleic acid biosynthesis, eventually killing pathogens. Two strategies are classically exploited: the inhibition of purines and pyrimidines biosynthesis or the impairment of nucleic acids polymerization. Recently, novel strategies have emerged, such as covalently binding the DNA to mechanically impair the enzyme binding. This is, for instance, the mechanism of action of the 5-Nitroimidazoles [88].

### 3.5.1. Inhibitors of DNA Gyrase

The DNA gyrase is a peculiar type IIA topoisomerase involved in the topology of DNA and a validated druggable target in mycobacteria [89]. This enzyme is a hetero-tetramer composed of two GyrA subunits and two GyrB subunits ($A_2B_2$ complex). In the presence of magnesium and upon ATP hydrolysis, it introduces a nick in the double strand to resolve the supercoil during DNA replication and mRNA synthesis. It is also involved in maintaining the homeostasis of negative supercoil DNA [89].

Tricyclic pyrrolopyrimidine (TPP) and SPR719 are inhibitors of the *Mtb* gyrase. They bind the ATPase domain of the B subunit, impairing the enzymatic activity [90]. TPP8, shown in Table 6, is a derivative of the TPP scaffold that exhibited a MIC in the range of 0.2–0.02 μM against various *Mab* strains, ATCC19977 included, and a panel of clinical isolates. In vitro DNA supercoiling inhibition studies confirmed that TPP8 impaired the enzymatic activity with an $IC_{50}$ of 0.3 μM. Finally, the two resistant mutants isolated in the study have been analyzed by whole-genome sequencing. They had a C506A (Thr169Asn) missense mutation in the ATPase domain, confirming the TPP8 mechanism of action [91]. This molecule showed a bacteriostatic effect against *Mab* in a THP-1-derived macrophage infection model treated with a different MIC multiple.

Similarly, SPR719 (Table 6) targets the ATPase domain of GyrB, acting as a competitive inhibitor of ATP [92]. Its mechanism of action has been identified through the isolation of *Mtb*-resistant mutants and their genetic analysis. The oral formulation of this molecule already passed the phase 1 clinical trial for safety, tolerability, and pharmacokinetics [93]. A recent evaluation of this molecule against *Mab* ATCC19977 and *Mycobacterium avium* showed MIC values of 1.6 μM and 6 μM, respectively. Additionally, SPR719 lacks ex vivo and in vivo experiments to further confirm its efficacy against *Mab*-infecting cells or animal models.

**Table 6.** Structures of molecules that target *Mab*'s nucleic acids biosynthesis.

| Code | Structure | Code | Structure |
|---|---|---|---|
| TPP8 |  | SPR719 |  |
| 844-TMF |  | EC/11716 |  |
| MMV688845 |  | Ryfamicin O |  |

Piperidine-4-carboxamides are a class of molecules that inhibit mycobacterial gyrase in a different way than TPPs and SPR719. Indeed, resistant mutants suggest that this scaffold targets GyrA instead of GyrB [94]. Two derivatives of this scaffold, molecule 844 and 844-TMF (Table 6), were active against the three reference strains of the *Mab* complex, and a collection of clinical isolates showing MICs in the range of 6–14 µM for 844 and 1.5–6 µM for 844-TMF. Moreover, both of the molecules had a bactericidal effect on planktonic cultures and biofilm [94]. Despite the positive results of these molecules and their interesting pharmacokinetic properties, it is necessary to further evaluate this molecule to confirm its efficacy in vivo.

A different mechanism of action is explored by the *Mab* gyrase inhibitor EC/11716 (Table 6). This molecule is classified as part of "novel bacterial topoisomerase inhibitors". This class of compounds is characterized by a two-part scaffold, the right-hand side, and the left-hand side, linked together by a central unit. These molecules impair the gyrase enzymatic activity by binding a non-catalytic interface pocket between the two GyrA subunits on the right-hand side and intercalating the DNA after cleavage with its left-hand side [95]. EC/11716 was evaluated against *Mab* ATCC19977 and a pool of clinical isolates showing MICs in the order of a few micromolar against every strain, with a bactericidal effect against planktonic cultures and impairing biofilm formation [96]. Moreover, it decreases 1 log of the bacterial CFU of *Mab* K21 at a daily dose of 400 mg/Kg after 10 days in the NOD SCID mice infection model, confirming the potentiality of this molecule in vivo.

### 3.5.2. Inhibitors of Polymerases

Among classical targets of DNA metabolism, DNA polymerase plays a major role. In particular, DNA polymerase III β subunit (DnaN) is involved in DNA replication and repair in bacteria. DnaN is a sliding clamp that acts as protein–protein interaction (PPI) hub by surrounding the double-strand DNA and recruiting a plethora of proteins involved in DNA metabolism [97,98].

Griselimycins are a class of cyclic depsipeptides originally isolated by *Streptomyces* spp. [99], which binds the PPI cleft, impairing the interaction with the polymerase III α subunit DnaE1 [100]. The cyclohexyl-griselimycins (CGM) analog showed a potent antitubercular effect in vivo in infected TB mouse models and good oral bioavailability [100]. Recently, CGM was evaluated against three reference strains of *Mab*, ATCC19977, *Mab* subsp. *Massieliense*, and *Mab* subsp. *bolletii* and against a pool of clinical isolates showing

positive results. CGM exhibited MICs against all the considered strains ranging between 0.1 and 0.8 μM with a bactericidal effect in all cases. It was also effective in vivo, decreasing the *Mab* K21 CFU by 10-fold compared to the same dose of CLA in an infected mouse model [101].

Nα-2-thiophenoyl-d-phenylalanine-2 morpholinoanilide (MMV688845), shown in Table 6, is an interesting inhibitor of RNA polymerase [102]. Its mechanism of action was verified by analysis of six isolated *Mab*-resistant mutants and by analogy with other inhibitors of the RpoB subunit of the *Mtb* RNA polymerase sharing the same chemical scaffold [103]. MMV688845 inhibited the growth of different *Mab* strains in vitro with a MIC in the order of a few μM and ex vivo showing a MIC of 15.9 μM in a macrophage infection model, exhibiting bactericidal effect [102]. Moreover, it showed synergy in combination with CLA, lowering the MIC by 10-fold compared to other antibiotics [102]. Despite these positive results, in vivo studies on mice demonstrated that the plasm concentration of the oral formulation of this compound never reaches the MIC value necessary to eradicate the infection [102], so the improvement of pharmacokinetic properties is required.

RpoB is also targeted by rifamycin, but *Mab* is naturally resistant to this drug [104]. Rifamycin O (Table 6), the oxidated derivative of the natural rifamycin B obtained by *Streptomyces mediterranei* [105], showed interesting results even against this microorganism. Indeed, the in vitro evaluation against *Mab* CIP 104536[T], *Mab* subsp. *bolletii* CIP108541[T] and *Mab* subsp. *massiliense* CIP108297[T] showed a $MIC_{90}$ of 4.0–6.2 μM. Moreover, a concentration of 25 μM of rifamycin O daily used to treat a zebrafish infection model showed a reduction of *Mab* infection and the increase of zebrafish lifespan of 54% up to 13 days, proving that the molecule is effective in vivo and confirming its low cytotoxicity [104].

### 3.6. Inhibitors of Protein Biosynthesis

Protein biosynthesis is an essential metabolic mechanism that requires the coordination of many players, such as ribosomes, mRNA, tRNA, and other factors, such as initiation factors IF and elongation factors EF [106–108]. The impairment of one of these players blocks protein synthesis; hence, many drugs have been developed exploiting this strategy.

### 3.6.1. Molecules That Directly Target the tRNA

MRX-6038 is a boron-containing molecule that inhibits the prokaryotic leucyl-tRNA synthetase (LeuRS). This class of inhibitors covalently binds the tRNA at the terminal adenine 76 of the editing site, thereby hindering leucylation and consequentially impairing protein synthesis [109]. MRX-6038 mechanism of action was validated by sequencing resistant mutants, which exhibited different mutations in the editing site [110]. MRX-6038 was evaluated against two *Mab* reference strains, ATCC19977 and CIP108297, along with 194 clinical isolates. The MICs for all the considered strains ranged between 0.063 and 0.125 mg/L [110]. Furthermore, it was assessed against 10 NTM reference strains, exhibiting MICs below 0.25 mg/L against *Mycobacterium smegmatis*, *Mycobacterium fortuitum*, *Mycobacterium scrofulaceum*, and *Mycobacterium peregrinum*, and other 33 NTM clinical isolates showing comparable MIC values. Moreover, it showed bactericidal or bacteriostatic effects in a range between 2 and 16 mg/L, depending on the strain. In the J774A.1 macrophage infection model, MRX-6038 reduced the intracellular *Mab* growth in a dose-dependent manner and showed no cytotoxicity up to 4-fold the MIC. In the mouse lung infection model, a daily administration dose of 10 mg/kg of MRX-6038 reduced *Mab* infection with a similar effect as LIN and CLA [110].

Other compounds characterized by the benzoxaborole scaffold targeting LeuRS are the nonhalogenated benzocaboroles, such as the so-called epetraborole. Part of this class of compound is EC/11770, a molecule effective against both *Mtb* and *Mab* [111]. Its structure is shown in Table 7. EC/11770 mechanism of action was confirmed in *Mab* by isolation of resistant mutant [111]. Indeed, it showed a MIC against the clinical isolate *Mab* Bamboo of 1.2 μM and 0.7 μM in 7H9 and CAMH media, respectively. It was evaluated against all the strains of the *Mab* complex and a panel of clinical isolates exhibiting MICs comparable to

Bamboo [111]. EC/11770 was previously defined as bactericidal in *Mtb.* This effect was confirmed in *Mab* exhibiting a minimal bactericidal concentration (MBC) of 50 µM against the biofilm and >100 µM against *Mab* planktonic cells. Nonetheless, since the MIC for the biofilm was 3.1 µM and for planktonic cells 3.0 µM, these data confirm the bacteriostatic effect of this molecule [111]. The effectiveness of this compound was confirmed in vivo in experiments on the murine model, resulting in a decrease of CFU both in the lungs and spleen after the administration of 300 mg/kg [112].

**Table 7.** Structures of a selection of inhibitors of *Mab* proteins biosynthesis.

| Code | Structure | Code | Structure |
|---|---|---|---|
| EC/11770 |  | AW6 |  |
| RUFI |  | Compound 12 |  |

### 3.6.2. Molecules That Target the Enzymes

Another different tRNA target is the tRNA-(N(1)G37) methyltransferase TrmD. Its role is to avoid +1 translational frameshift errors during protein biosynthesis in prokaryotes. It belongs to the class of the S-adenosyl-L-methionine (SAM)-dependent methyltransferases (Class IV methyltransferases), whereas the eukaryotic counterpart Trm5 belongs to Class I methyltransferases. This difference allows for the exploitation of this protein as a target in bacteria since the putative inhibitors would not target the eukaryotic orthologue [113]. Moreover, the essentiality of TrmD in *Mab* has been confirmed by mutagenesis experiments, homologous recombination, and allelic replacements of the *trmD* gene [114]. Some inhibitors of this enzyme have been developed through the so-called fragment-based drug discovery (FBDD). A huge library of small molecules has been screened for the identification of some structures that may interact with the SAM binding pocket of TrmD. The best candidates, which could bind the target in different positions, can be implemented or linked together to improve their potency. In this study, the most effective fragments were merged into the AW1 compound, which exhibited a $K_d$ 0.11 mM and $IC_{50}$ 0.23 mM against the enzyme [114]. Further improvements of AW1 resulted in three more effective derivatives, AW5 ($K_d$ 27 nM, $IC_{50}$ 30 nM), AW6 ($K_d$ 0.49 µM, $IC_{50}$ 1.4 µM), and AW7 ($K_d$ 73 nM, $IC_{50}$ 69 nM), which exhibited MICs of 50 µM each. Moreover, AW6 (Table 7) and AW7 were evaluated ex vivo, showing a decrease in *Mab* CFU in an infected human macrophage model at a concentration of 25 µM of ~82% and 95%, respectively, without cytotoxic effect up to 150 µM. The specificity of the target has been confirmed by *trmD* silencing, which resulted in a proportional susceptibility of *Mab* to AW7 following the increasing silencing of the protein expression [114].

Tetracyclines are known inhibitors of bacterial ribosomes [115]. New derivatives of this class, such as the aminomethylcycline omadacycline [116], the halogenated tetracycline eravacycline [117], and the glycylcycline tigecycline [118], have been evaluated against *Mab* ATCC19977 and 28 drug-resistant *Mab* clinical isolates, showing MICs of ≤4 µg/mL [119–121]. Although omadacycline seems to have a bacteriostatic effect [120] even at higher concentrations, preliminary pharmacokinetics and dynamics experiments

in a static drug concentration scenario showed better behavior compared to tigecycline, suggesting that this molecule could be better tolerated by the human body [119]. Moreover, the aminomethyl substituent at the C-9 position on the tetracycline D-ring allows omadacycline to bypass the common resistant mechanisms to other tetracyclines [121].

The ATP-dependent caseinolytic protein C1 (ClpC1) is an emerging druggable target among the ClpC chaperone proteins [122]. It is involved in the protein degradation processes in complexes with ClpC2. In particular, the importance of ClpC1 rather than ClpC2 stands in its capability of regulating the proteolytic domains of both enzymes [122]. Two mechanisms have been proposed to perturbate this complex: uncoupling the enzymes or overactivating ClpC1 to induce intracellular uncontrolled protein degradation. The cyclic heptapeptide rufomycins (RUFs), also known as ilamycins and originally obtained from *Streptomyces atratus* and *Streptomyces macrosporeus* [123], are a class of compounds that target ClpC1. The RUFI derivative, as shown in Table 7, has been well-characterized against *Mtb* MDR and XDR strains. The repurposing of this molecule showed selective bactericidal activity against *Mab*, exhibiting a MIC of 0.42 μM and MBC at 1.2 μM. It was also effective in the bone-marrow macrophage infection model, showing a comparable effect to CLR [122]. Isolation of resistant mutants to RUFI showed one to seven mutations on the N-terminal domain of ClpC1, suggesting that this compound may bind the enzyme in this location [122]. Surface plasmon resonance studies on RUFI proved that it targets *Mab* ClpC1, impairing the proteolytic activity of the complex ClpC1C2 and leaving unaltered the ClpC1 ATPase activity [122]. Apart from RUFI, other RUF analogues showed MICs in the order of a few micromolar. For example, compounds 7, 8, and 12 (Table 7) exhibited MICs against *Mab* of 0.54 μM, 0.58 μM, and 0.29 μM, respectively [123]. In contrast, compounds 4 and 9 showed worse MICs of >10 μM, suggesting a general high specificity of this class of compound for ClpC1 [123].

### 3.7. Inhibitors of Coenzyme A Biosynthesis

Mycobacterial L-aspartate α-decarboxylase, PanD, is involved in the coenzyme A (CoA) biosynthesis. It catalyzes the condensation of β-alanine and D-pantoate into pantothenate, a CoA precursor, upon conversion of ATP into ADP [124]. Since the absence of this pantothenate synthase in humans and the essentiality of CoA in all organisms, the enzymes involved in the CoA biosynthetic pathway are interesting drug targets against bacteria. Pyrazinamide (PZA) is a prodrug activated by the pyrazinamide (PncA) into the pyranozoic acid (POA). POA inhibits *Mtb* PanD, but it is not active against the *Mab* homologous enzyme, exhibiting only 14% inhibition at 200 μM through β-alanine production assay [125]. Recently, some POA analogs have been evaluated against *Mab* PanD. Of these, compound 2 resulted in the best inhibition of 72% and $IC_{50}$ of 55 μM, while other analogs exhibited a milder effect. Anyway, the most effective molecules of this series were characterized by the 3-arylamido substituent in position 3 of POA, whereas other substituents, such as amino, cyclo-alkilamino, benzylamino, and aryl group, resulted in insufficient inhibition percentage of the β-Ala production [125]. Despite the high values of $IC_{50}$ of compound 2 compared to the best *MbtI* PanD inhibitors, data suggest that this molecule is a promising starting point for designing new drugs against *Mab*. This assumption is supported by its $MIC_{50}$ of 0.7 mM against *Mab* ATCC19977, comparable to the effect of POA against *Mtb*, and the MICs ranging between 1 and 1.2 mM assessed against the other *Mab* species of the complex and against the clinical isolate Bamboo [125].

A different target of mycobacterial CoA biosynthesis is the phosphopantetheine adenylyltransferase (PPAT or CoaD) enzyme involved in the penultimate step of this biosynthetic pathway. A structure-guided fragment-based drug discovery over *Mab* PPAT led to the identification of 16 potential drug fragments. Besides these hits, fragments already obtained from fragment-based analysis against *Mtb* PPAT have also been analyzed against *Mab*. From the first selection, only fragments 5 and 20 (Table 8) had a good dissociation constant against *Mab* PPAT, inducing important conformational changes in its structure. While compound 5 bound the sub-pocket II of the enzyme with a $K_d$ of 3.2 ± 0.02 mM,

compound 20 bound the ATP binding site with a $K_d$ of $0.54 \pm 0.02$ mM [126]. However, these molecules need to be further developed since they have a weak effect in vitro against *Mab*. Nonetheless, the parallel studies focused on the binding between the fragments and the enzyme may help the design of effective anti-mycobacterial drug compounds [126].

**Table 8.** Structures of two inhibitors of *Mab*'s phosphopantetheine adenylyltransferase.

| Code | Structure | Code | Structure |
|------|-----------|------|-----------|
| Fragment 5 |  | Fragment 20 |  |

### 3.8. Targeting Virulence Factors

An emerging strategy to kill bacteria is targeting virulence factors. Indeed, inhibition of non-essential pathways involved in bacterial growth or disruption of the processes involved in the pathogenesis applies a low evolutive selection compared to targeting essential genes [127]. For instance, targeting the biosynthesis of mycobacterial siderophores, known as mycobactins and involved in host iron scavenging, is an exquisite strategy attracting different study groups of *Mtb*. In *Mab*, the first enzyme involved in the mycobactins biosynthesis is the salicylate synthase SaS, which converts the chorismate into salicylate [128]. Since it is homologous in *Mtb*, the MbtI enzyme has been widely explored [129–131], and a library of putative inhibitors has been recently evaluated against *Mab*-SaS, considering the structural and functional similarities shared by these two proteins. This library was composed of 5-phenylfuran-2-carboxylic acid-derived molecules. Three compounds showed $IC_{50}$s ranging between 5.3–29.5 μM. Despite the 5-(2,4-bis(trifluoromethyl)phenyl) furan-2-carboxylic acid (compound 1, Table 9) assessed an $IC_{50}$ of $5.3 \pm 1.5$ μM against the enzyme and decreased the production of the siderophore in a concentration-dependent manner, it did not show any toxic effect against *Mab* in liquid culture and iron-limiting condition up to 500 μM [128]. This preliminary data opens the path to new possibilities for the development of effective drugs against *Mab*.

**Table 9.** Structures of two inhibitors of *Mab*'s virulence factors.

| Code | Structure | Code | Structure |
|------|-----------|------|-----------|
| Compound 1 |  | GaPP |  |

Other approaches to exploit the mycobacterial iron metabolism as a drug target are ongoing, like mimicking the iron's physical–chemical properties with other metals that can compete for the binding with mycobactins. Iron $Fe^{3+}$ and gallium $Ga^{3+}$ have indeed similar features; indeed, gallium-based molecules have been successfully applied to fight other bacterial infections [132–134]. Gallium has the ability to disrupt the iron acquisition systems and to impair the function of heme-utilizing hemeproteins; hence, it perturbates multiple aspects of pathogens [127].

Gallium protoporphyrin (GaPP, Table 9), gallium mesoporphyrin (GaMP), and gallium nitrate ($Ga(NO_3)_3$) have been tested against *Mab* in a depleted-iron medium. $Ga(NO_3)_3$

showed a MIC of 16 μg/mL, while better results were obtained with porphyrins: the MIC of GaPP was 0.5 μg/mL and GaMP of 1 μg/mL. Moreover, some compound combinations have been evaluated through checkerboard assay to verify a potential synergistic effect. Fractional inhibitory concentrations (FIC) resulted in 0.04 and 0.06 for GaMP/Ga(NO$_3$)$_3$ and GaPP/Ga(NO$_3$)$_3$, respectively, confirming the synergy for these two combinations since the values are $\leq$0.5 [135]. GaPP has also been evaluated in combination with first-line drugs, resulting in FICs ranging from 0.28 to 0.5, while Ga(NO$_3$)$_3$ in combination with first-line drugs resulted in a mean FIC of 0.5. Despite the positive data of these combinations, they are not comparable with the synergistic effect between Ga-porphyrins and Ga-nitrate. Further experiments against the R morphotype of *Mab* were conducted in the infected mice model. In this context, the combination of Ga-based compounds inhibited the *Mab* aconitase enzyme at 1$\times$ and 10$\times$ concentrations, whereas 1$\times$ was 0.5 μg/mL for Ga(NO$_3$)$_3$ and 0.016 μg/mL for GaPP. Moreover, they showed a mild effect against the *Mab* catalase, but no synergy has been observed against this enzyme. Finally, the synergistic effect of the Ga(NO$_3$)$_3$/GaPP combination showed a significant reduction of *Mab* infection in mice at two different concentrations, resulting in 100% alive mice at 10x concentration and 60% of alive mice at 1X. The bacterial load in both cases decreased significantly thanks to the synergistic effect compared to the mice population treated with the single compound [135].

### 3.9. Unknown Targets

Different molecules are under investigation against *Mab*, but their mechanism of action is still unknown.

Etamycin has shown good efficacy in vitro against different strains and clinical isolates, with MICs ranging between 28.3 and 4.3 μM. Furthermore, it exhibited efficacy against both R and S morphotypes within a similar range. Ex vivo experiments conducted on a macrophage infection model and in vivo experiments on zebrafish confirmed etamycin efficacy in a dose-dependent manner in both models. Interestingly, no cytotoxic effect was observed on various human cell lines up to the concentration of 50 μM [136].

The 10-DEBC is a molecule structurally related to chlorpromazine (Table 10), a phenothiazine compound known for its effectiveness against different drug-resistant strains of *Mtb* [137]. Although its mechanism of action is not well understood, it has been hypothesized that 10-DEBC may interact with multiple targets in both *Mtb* and the host cell. Notably, it has been demonstrated that 10-DEBC targets Akt, a serine/threonine kinase involved in the regulation of apoptosis in macrophages. Inhibition of Atk promotes cell survival [138]. The 10-DEBC has been tested in vitro against *M. abscessus*-LuxG13, exhibiting a comparable efficacy to CLR, AMK, and STR, with a MIC of 12.5 μg/mL. Additionally, it displayed a lower MIC against nine other clinical isolates, ranging from 2.38 to 4.77 μg/mL [139]. The same study proved its efficacy against non-replicating *Mab* under anaerobic conditions and biofilm-growing *Mab* with a MIC of 50 μg/mL [139]. The 10-DEBC-inhibited *Mab* exhibited growth in two infection models: in THP-1 monocytes showing an IC$_{50}$ of 13.18 μg/mL, and in human embryonic cell-derived macrophages iMACs, with an IC$_{50}$ of 3.48 μg/mL [139]. Despite the limited understanding of the 10-DEBC mechanism of action, data suggest it could be a valuable addition to the current therapeutic regimen.

**Table 10.** Structures of 10-DEBC, one inhibitor of *Mab* growth. Its molecular target is unknown.

| Code | Structure |
|---|---|
| 10-DEBC |  |

## 4. Conclusions

As previously discussed, the majority of drugs employed to treat *Mab* infections are mostly repurposed and initially identified against *Mtb*, such as BDQ, AMK, azithromycin, and tigecyline. However, *Mab*'s propensity to develop resistance drives the research for alternative options. Therefore, the modification of existing drugs in order to enhance their efficacy against *Mab* is combined with the development of completely new molecules. For instance, derivatives of ICs, such as IC5 and IC25, showed promising results against this microorganism.

Ongoing research considers all the possible targets, such as membrane biosynthesis, essential enzymes, and virulence factors. As a result, different drugs have been designed to target *Mab* with high efficacy. This is proved by MICs in the order of a few micromolar and by positive results in in vivo experiments. In the near future, it is possible to expect that a deeper understanding of *Mab* features associated with a better knowledge of drug targets and the molecular mechanism of action of molecules in development will lead to the improvement of more effective therapies to fight this pathogen.

**Author Contributions:** Conceptualization, G.S.; writing—original draft preparation, M.C., C.B., S.F. and G.S.; writing—review and editing, G.S.; visualization, M.C. and G.S. All authors have read and agreed to the published version of the manuscript.

**Funding:** This research received no external funding.

**Institutional Review Board Statement:** Not applicable.

**Informed Consent Statement:** Not applicable.

**Data Availability Statement:** Not applicable.

**Conflicts of Interest:** The authors declare no conflict of interest.

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
