# Peer review of "Moles of Molecules against Mycobacterium abscessus: A Review of Current Research"

_futurepharmacol, doi:10.3390/futurepharmacol3030041_

Round 1

Reviewer 1 Report

This is an interesting review focusing on drug therapies under investigation for one of the atypical mycobacteria, M. abscessus. Atypical mycobacterial disease is a rising problem in clinical practice since there are an increasing number of immunocompromised patients which are one of the main risk groups of the infection. The paper is well written and all the necessary references have been included. One minor observation: the authors could also add a table in their paper listing all the drugs which are mentioned in detail in their review. Also there are treatment guidelines for non-tuberculous mycobacterial pulmonary disease which must be included in the paper. 

The paper is well written, minor editing is required. 

Author Response

We want to thank the reviewer for the nice evaluation and the valuable comments.

Reviewer: One minor observation: the authors could also add a table in their paper listing all the drugs which are mentioned in detail in their review. Also there are treatment guidelines for non-tuberculous mycobacterial pulmonary disease which must be included in the paper. 

Reply: According to the reviewer valuable suggestions, we added two tables to summarize and classify better all the treated molecules. We also added a sentence about the guidelines for NTM (line 76-79). This sentence is based on clinical article published last year: https://doi.org/10.1016/j.ccm.2021.11.007.

Reviewer 2 Report

General Comments

1. A timely and important contribution, but difficult to read as there are no sub-headings and paragraph titles.

2. Specific Comments

Title: What is the purpose of "Moles", its inclusion is unclear.

Abstract

Lines 13-14. Most M. abscessus drugs have come from those used against nontuberculous mycobacteria, namely M. avium, and not from TB.

ntroduction

Lines 58-60. Is the change from S to R, "irreversible" or can R types revert to S? Further, rather than a review, please cite specific S to R colony variation paper.

Lines 61-65. This paragraph does not make sense. Specifically, "infections", the subject of the sentence, are not involved in the "formation" of the M. abscessus complex. I suggest simply deleting the whole paragraph, or based on the authors review, to address whether it is necessary to develop drugs against the other members of the M. abscessus complex.

Section 2. "Ongoing...", Lines 79-end. Second 2 provides an unorganized and untitled listing of drugs and the status of their individual investigations. Because of the lack of paragraph headings, it is difficult to follow the information. Please consider overlaying the individual drug stories with headings to highlight the major points and provide ready access to specific drugs.

Author Response

We thank the reviewer for his attention in reviewing our article and the valuable comments. Following our reply.

Reviewer: 1. A timely and important contribution, but difficult to read as there are no sub-headings and paragraph titles.

Reply: We understand your concern, overall for the initial part of the article. We just followed the guidelines of MDPI reported in the template. Although, we are afraid that further subdivisions would make desultory the text.

Reviewer: Title: What is the purpose of "Moles", its inclusion is unclear

Reply: it is just a word joke, it gives a nice sound to the sentence thanks to the assonance with “molecules”.

Reviewer: Lines 13-14. Most M. abscessus drugs have come from those used against nontuberculous mycobacteria, namely M. avium, and not from TB.

Reply: The sentence in abstract is due to the fact that the molecules described in the review are mostly repurposed by TB. Indeed, if you checked the description of the drugs that are currently in clinical phases or early drug clinical phase, you can immediately identify how many drugs have been repurposed from TB treatment. For what I read in literature, it seems that even some drugs used against M. avium are repurposed from tuberculosis (such as Rifampicin, Clarithromycin and Ethambutol – DOI: 10.1016/j.tube.2018.12.004), this is mostly due to the fact that tuberculosis is the major concern and most studied mycobacteria.  Hope of not being offensive by providing you a small list of articles which describes the identification and repurposing of anti-TB drugs for Mab as example:

  • DOI:1183/16000617.0212-2021
  • DOI: 10.3390/antibiotics9010018
  • DOI: 10.1128/aac.02420-20
  • DOI: 10.1016/j.drudis.2018.04.001
  • DOI: 10.1128/aac.00363-20
  • DOI: 10.1080/17460441.2019.1629414

Reviewer: Lines 58-60. Is the change from S to R, "irreversible" or can R types revert to S? Further, rather than a review, please cite specific S to R colony variation paper.

Reply: Thank you for the comment. The change from S to R is indeed irreversible and depends on genetic modifications. We added the reference of the more informative original article instead that the review, as suggested.

Reviewer: Lines 61-65. This paragraph does not make sense. Specifically, "infections", the subject of the sentence, are not involved in the "formation" of the M. abscessus complex. I suggest simply deleting the whole paragraph, or based on the authors review, to address whether it is necessary to develop drugs against the other members of the M. abscessus complex.

Reply: We thank the reviewer for the valuable comment. We modified the sentence to clarify the message (line 55-59).

Reviewer: Section 2. "Ongoing...", Lines 79-end. Second 2 provides an unorganized and untitled listing of drugs and the status of their individual investigations. Because of the lack of paragraph headings, it is difficult to follow the information. Please consider overlaying the individual drug stories with headings to highlight the major points and provide ready access to specific drugs.

Reply: We understand the reviewer concern. We had the same doubts about this section, and in general about the organization of the whole manuscript. At the end, following MDPI template for the sections and sub-sections we decided to describe every drug in a paragraph and to divide drugs according to the clinical phase in evaluation. In this way is possible to identify the space dedicated to a specific molecule, as well as where it starts and finishes. We took this decision since there are only few drugs in clinical phases evaluation, mostly with a different target. We are afraid that other modifications in this section would make desultory the text.

Reviewer 3 Report

Moles of molecules against Mycobacterium abscessus - A
 review of current research.

Dear author and editor:

this review article talked could be published in future pharmacology after a minor revision

I have some comments:

·        What do you think about host directed therapy against Mycobacterium and Mycobacterium abscessus.

·        What do you think about non-antibiotic therapy against Mycobacteium abscessus.

Thank you very much, best regards

Author Response

We want to thank the reviewer for his kind work. Following our reply.

Reviewer: What do you think about host directed therapy against Mycobacterium and Mycobacterium abscessus.

Reply: According to our work we have not direct clinical relationships with patients so we didn’t take host-directed therapy into account. Anyway, I think that your comments is a valid implementation for our manuscript. For this reason we added a sentence that may recall the topic (line 97-99).

Reviewer: What do you think about non-antibiotic therapy against Mycobacteium abscessus.

Reply: The reviewer touched a very good point. Writing the manuscript we intentionally didn’t want to discuss the non-antibiotic therapies to not implement off-topics, since the review is already long and intense. Anyway, we wanted to meet the reviewer comment by briefly implementing the paragraph in lines 92-96.

Thank you for your valuable comments, they really helped us to improve our manuscript.